# Self-Assembled Monolayer of Monomercaptoundecahydro-*closo*-dodecaborate on a Polycrystalline Gold Surface

**DOI:** 10.3390/molecules27082496

**Published:** 2022-04-12

**Authors:** Martha L. Jiménez-González, Juan Pablo F. Rebolledo-Chávez, Marisela Cruz-Ramírez, René Antaño, Angel Mendoza, Narayan S. Hosmane, Lena Ruiz-Azuara, José Luis Hernández-López, Luis Ortiz-Frade

**Affiliations:** 1Centro de Investigación y Desarrollo Tecnológico en Electroquímica S.C, Parque Tecnológico Querétaro, Querétaro 76703, Mexico; mjimenez@cideteq.mx (M.L.J.-G.); rantano@cideteq.mx (R.A.); 2División de Química y Energías Renovables, Universidad Tecnológica de San Juan del Río, Avenida La Palma No. 125 Vista Hermosa, Querétaro 76800, Mexico; jprebolledoc@utsjr.edu.mx (J.P.F.R.-C.); mcruzr@utsjr.edu.mx (M.C.-R.); 3Centro de Química, Instituto de Ciencias, Benemérita Universidad Autónoma de Puebla, Ciudad Universitaria, Col. San Manuel, Puebla 72570, Mexico; angel.mendoza@correo.buap.mx; 4Department of Chemistry and Biochemistry, Northern Illinois University, DeKalb, IL 60115, USA; hosmane@niu.edu; 5Facultad de Química, Universidad Nacional Autónoma de México, Mexico City 04510, Mexico; lenar701@gmail.com; 6Centro de Nanociencias y Micro y Nanotecnologías del Instituto Politécnico Nacional, Luis Enrique Erro s/n, Unidad Adolfo López Meteos, San Pedro Zacatenco, Alcaldía Gustavo A. Madero, Mexico City 07738, Mexico

**Keywords:** monomercaptoundecahydro-*closo*-dodecaborate, self-assembled monolayer, cyclic voltammetry, electrochemical impedance spectroscopy, surface plasmon resonance

## Abstract

In this work, we present an electrochemical study of the boron cage monomercaptoundecahydro-*closo*-dodecaborate [B_12_H_11_SH]^2−^ in solution and in a self-assembled monolayer over a polycrystalline gold electrode. Cyclic voltammetry of the anion [B_12_H_11_SH]^2−^ in solution showed a shift in the peak potentials related to the redox processes of gold hydroxides, which evidences the interaction between the boron cage and the gold surface. For an Au electrode modified with the anion [B_12_H_11_SH]^2−^, cyclic voltammetry response of the probe Fe(CN)_6_^3−^/Fe(CN)_6_^4−^ showed a ΔEp value typical for a surface modification. Electrochemical impedance spectroscopy presented *R_tc_* and C_dl_ values related to the formation of a self-assembled monolayer (SAM). A comparison of electrochemical responses of a modified electrode with thioglycolic acid (TGA) reveals that the boron cage [B_12_H_11_SH]^2−^ diminishes the actives sites over the Au surface due to the steric effects. Differential capacitance measurements for bare gold electrode and those modified with [B_12_H_11_SH]^2−^ and (TGA), indicate that bulky thiols enhance charge accumulation at the electrode–solution interface. In addition to electrochemical experiments, DFT calculations and surface plasmon resonance measurements (SPR) were carried out to obtain quantum chemical descriptors and to evaluate the molecular length and the dielectric constant of the Boron cage. From SPR experiments, the adsorption kinetics of [B_12_H_11_SH]^2−^ were studied. The data fit for a Langmuir kinetic equation, typical for the formation of a monolayer.

## 1. Introduction

The use of the boron cage monomercaptoundecahydro-closo-dodecaborate [B_12_H_11_SH]^2−^ (Figure 1) as a chemotherapeutic agent for cancer treatment [1,2,3] has motivated chemists to study their reactions with proteins, metal clusters and different organic molecules in order to improve the biological activity for adequate drug delivery [4,5,6,7,8].

Conversely, boron cages such as [B_12_H_11_SH]^2−^ have been used to understand the steric, geometric, and electronic factors that control the self-assembly of monolayers. Particularly, Cs_2_[B_12_H_11_SH] was the first example of a dodecacarborane derivative used in a self-assembly with an Au surface, studied by surface-enhanced Raman spectroscopy (SERS); however, the negative charges and the presence of cesium atoms complicate the imaging and structural determination by this technique [9,10].

Furthermore, the formation of thiol self-assembled monolayers on gold surfaces has been the subject of many studies because they offer different surfaces with controlled chemical or physical properties [11], having an enormous impact in the field, such as corrosion, lubrication, biomimetics and sensing, among others [12]. However, the study of SAM over Au electrodes with bulky molecules has not been explored extensively in comparison with alkanethiol derivatives.

The aforementioned motivated us to study the electrochemical response of the boron cage [B_12_H_11_SH]^2−^, Figure 1, in solution and in a self-assembled monolayer over a polycrystalline gold electrode, using cyclic voltammetry, electrochemical impedance spectroscopy and differential capacitance, in order to explore the effect of bulky molecules in the formation of SAM over Au electrodes. In combination with electrochemical experiments, DFT calculations and surface plasmon resonance measurements (SPR) allow us to obtain quantum chemical descriptors and to determine the molecular length and the dielectric constant of the Boron cage. The results are compared with experiments obtained from the modification of gold electrodes with thioglycolic acid (TGA). SPR experiments were carried out to explore the kinetic model for the adsorption of the [B_12_H_11_SH]^2−^ over a gold surface. The results obtained in this work are intended to contribute to the design of the self-assembled monolayers for bulky molecules, particularly boron cages on metal surfaces, with potential applications.

## 2. Experimental Section

### 2.1. Chemicals

All chemicals and solvents were used as received and without further purification.

### 2.2. Monomercaptoundecahydro-closo-dodecaborate

The synthesis of Na_2_[B_12_H_11_SH] has been reported previously [13]. Elemental analysis was performed in a Fissons Instruments analyzer model EA 1108, Calcd. H, 5.50; S, 14.58. Found H, 5.52; S, 14.60. ^11^B-NMR experiments were carried out using a VARIAN Unity Inova spectrometer at 128.3 MHz. NaBH_4_ and CD_3_OD were used as reference and solvent, respectively. ^11^B-NMR data; δ_1_(−9.736, 1B), δ_2_ (−14.203, 5B), δ_3_ (−16.320, 5B), and δ_4_ (−19.669, 1B).

### 2.3. Gold Electrode Preparation

A polycrystalline gold electrode with a geometrical area of 0.02 cm^2^ was polished with a monocrystalline diamond suspension (1.0 μm), using a commercial polisher (Buehler) at 240 RPM. Possible organic and inorganic impurities from the first step were removed by immersion in chromic mixture and rinsed with deionized water and ethanol. Then, the electrode was manually polished with α Alumina (0.3 μm), rinsed with deionized water and placed in an ultrasonic bath. Electrochemical cleaning was also performed by introducing the gold electrode in 0.5 M H_2_SO_4_ solution, followed by imposing a constant potential value of −1200 mV vs. Ag/AgCl for 60 s. Finally, cycles between −350 and 1450 mV were performed at scan rates of 100 and 50 mV/s until the characteristic polycrystalline gold profile was obtained.

### 2.4. Electrode Modification

Afterward, the electrode was cleaned, which was immersed into a 10 mM Na_2_[B_12_H_11_SH] solution in the presence of 0.1 M phosphate buffer, pH 7.2, for 60 min. Then, the electrode was rinsed with distilled water and dried to be placed into an electrochemical cell containing 0.1 M phosphate buffer, pH 7.2, to record its electrochemical response. The same experiments were carried out using thioglycolic acid solutions in the same conditions. 

### 2.5. Electrochemical Experiments

Electrochemical experiments were performed using a Biologic SP-300 potentiostat/galvanostat. A typical three-electrode array was used: a gold disk as working electrode, platinum wire as counter-electrode, and Ag/AgCl as reference electrode. All solutions were bubbled with nitrogen prior to each measurement for 10 min. Cyclic voltammetry experiments of the boron cage in solution were carried out using 10 mM of Na_2_[B_12_H_11_SH] in the presence of 0.1 M phosphate buffer, pH 7.2. Cyclic voltammetry and electrochemical impedance spectroscopy experiments before and after electrode modification with [B_12_H_11_SH]^2−^ were carried out in the presence of 2.5 mM Fe(CN)_6_^3−^/Fe(CN)_6_^4−^ as a redox probe containing a solution of 0.1 M phosphate buffer, pH 7.2. The electrochemical impedance spectroscopy measurements were obtained by applying a DC potential corresponding to the open circuit potential with an amplitude of 5mV and a frequency ranging from 10,000 to 10 Hz. Data processing and fitting were carried out using EIS Spectrum Analyzer Version 1.b software. Differential capacitance curves (CD) were obtained before and after electrode modification in a solution of 0.1 M phosphate buffer, pH 7.2, using a potential sweep from −0.4 to 0.6 V with a frequency of 30 Hz and an amplitude of 10 mV.

### 2.6. Surface Plasmon Resonance Experiments

Surface plasmon resonance (SPR) measurements were carried out with a commercial instrument (NanoSPR, Model: 321, Chicago, IL, USA) operating in Kretschmann configuration [14] equipped with a Ga-As laser diode (λ = 670 nm) as the excitation source. The instrument was equipped with a flow cell and fluidic tubing coupled to a microperistaltic pump (Masterflex C/L, Radnor, PA, USA). The sensor chip surfaces (20 × 20 × 1 mm) were obtained from NanoSPR, USA, and were prepared by electron beam evaporation of an Au thin film (*ca.* 45 nm) on microscope slides with 5 nm Cr as the adhesive layer. The Au substrates were extensively rinsed in Milli-Q water, absolute ethanol, and dried with a stream of N_2_ gas prior to their modification. The optical thickness (*d*) of the film was evaluated by simulating the SPR angular scanning with *WINSPALL* version 3.01, which is a program based on Fresnel formalism. For the fitting process, the values of the *optical constants* provided in the vendor’s materials database for gold (*n_Au_* = 0.15 + 3.60*i*), chromium (*n*_Cr_ = 2.10 + 2.37*i*), prism and microscope slides (1.6160) were used. For the BSH^2−^ self-assembled monolayer, the refractive index (*n*) was assumed to be *n* = 1.50. The kinetic adsorption process of the self-assembled monolayer (SAM) was monitored in real-time and characterized using the tracking mode of the SPR angular scanning at the minimum SPR angle. A laminar flow regime was used throughout the assembly procedure. In a typical experiment, the sensor chip surface was first equilibrated with water for a few minutes to obtain a stable baseline. Then, 500 μL of Na_2_[B_12_H_11_SH] (10 mM) solution was injected into the flow cell and allowed to adsorb on the sensor chip surface as a function of time. After a rinsing step in pure water, a stable plateau was reached, indicating the formation of a stable monolayer. All experiments were performed at room temperature.

### 2.7. Theoretical Calculations

A full geometry optimization without symmetry constraints and frequency calculations of the boron cage and thioglycolic acid were performed starting from crystallographic data, using the hybrid density functional B3LYP [15,16,17] with a 6-31g(d) [18] basis set in gas phase. Optimized geometries of local minima were verified by the number of imaginary frequencies (which should be zero). Previous studies indicate that DFT results are accurate to describe stabilities, equilibrium geometries and frontier orbitals for B3LYP density functional and 6-31g(d) basis set combination [19]. All quantum chemical calculations were performed using Gaussian 09 [20].

## 3. Results

### 3.1. Cyclic Voltammetry Experiments

Figure 1a shows the cyclic voltammetry response of a bare Au electrode containing a solution 0.1 M phosphate buffer, pH 7.2. Two peaks, I_a_ and I_c_ at 0.363 and 0.236 V/Ag-AgCl, can be observed, related to the formation and to the reduction of gold hydroxides Au + 3OH^−^ → Au(OH)_3_ ↓ + 3e^−^, Au(OH)_3_ ↓ + 3e^−^ → Au +3OH^−^ over the electrode surface. When experiments were carried out in the presence of Na_2_[B_12_H_11_SH] in solution (Figure 1b), a shift in the oxidation and reduction peak potentials (I_a_* and I_c_*) was observed. This behavior can be associated with the interaction between the -SH group from the boron cage and Au(OH)_3_ ↓, as a first step for the electrode modification. The electrochemical response of the modified electrode in solution of 0.1 M phosphate buffer, pH 7.2, is presented in Figure 1c. The absence of gold hydroxides redox processes suggests the formation of a surface array [B_12_H_11_S-Au]^2−^. The decrease in the current values in the double layer capacitance region from −0.400 to 0.200 V V/Ag-AgCl in the modified electrode in comparison with the experiments for a bare Au electrode confirms this hypothesis. With the information mentioned before and according to the literature, the following mechanism can be proposed [21]:3[B_12_H_11_SH]^2−^ + Au(OH)_3_ ↓ → 3[B_12_H_11_S-Au]^2−^+ 3H_2_O

To understand the blocking effects of the [B_12_H_11_S-Au]^2−^ assembly, cyclic voltammetry experiments using a redox active probe were carried out. Figure 2a shows a typical cyclic voltammogram of a bare Au electrode, in the presence of the probe Fe(CN)_6_^3^^−^/Fe(CN)_6_^4^^−^, where reversible redox behavior (ΔEp = 0.060 V) was observed. When the Au electrode was modified with the boron cage, the electrochemical response of Fe(CN)_6_^3^^−^/Fe(CN)_6_^4^^−^ (Figure 2b), shows a decrease in the anodic and cathodic currents. Additionally, a ΔEp value of 0.294 V was observed, which indicates that the electrode was partially blocked. A comparison between the charge involved (integrated area under the peaks) in the absence and in the presence of the boron cage, for both anodic and cathodic signals, indicates a decrease of 82% and 27% for the anodic and cathodic domain, respectively. These changes are directly correlated to the number of active sites blocked by the presence of the [B_12_H_11_S-Au]^2−^ self-assembled layer.

**Figure 1 molecules-27-02496-f001:**
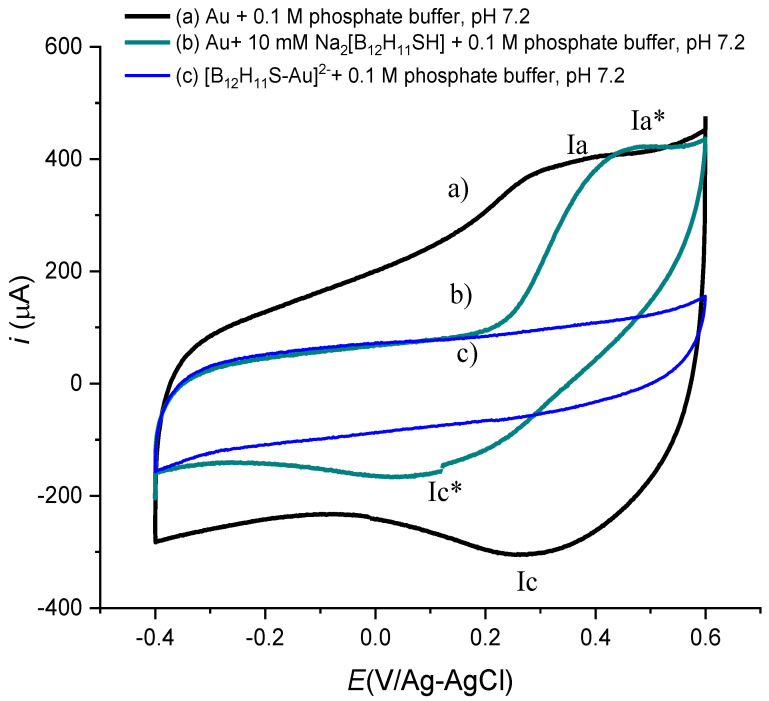
Cyclic voltammetry obtained with a Bare Au electrode (a) 10 mM of Na_2_[B_12_H_11_SH] in the presence of 0.1 M phosphate buffer, pH 7.2, scan rate 50 mVs^−1^ obtained with (b) bare Au electrode and (c) after modification with [B_12_H_11_SH]^2−^.

**Figure 2 molecules-27-02496-f002:**
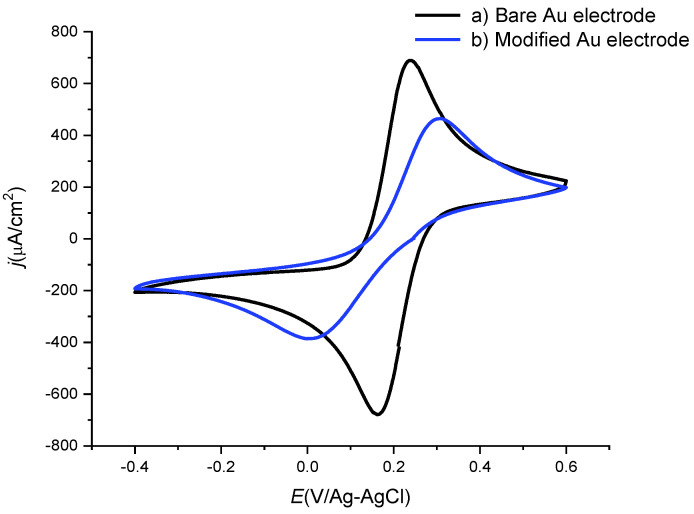
Cyclic voltammetry of 2.5 mM Fe(CN)_6_^3^^−^/Fe(CN)_6_^4^^−^ in the presence of 0.1 M phosphate buffer, pH 7.2, scan rate 50 mVs^−1^ obtained with (a) bare Au electrode and (b) after modification with [B_12_H_11_SH]^2−^.

Quantitative measurement of the blocking effect and dielectric properties of the modified Au interface can be obtained by means of electrochemical impedance spectroscopy, EIS.

### 3.2. Electrochemical Impedance Spectra of [B_12_H_11_S-Au]^2−^ Monolayer

Figure 3 shows a representative Nyquist diagram of a bare Au electrode and the same after its modification with the anion [B_12_H_11_SH]^2−^. It is evident that the spectra present a similar shape in both cases, as expected for the probe used [22]: a loop in the high frequencies domain, partially distorted by a linear trend in the low frequencies domain. The loop has been associated with the coupling of the double layer capacitance, C_dl_, with the charge transfer resistance, R_tc_, of the probe, whereas the linear trace is related to the diffusion in a semi-infinite state of the probe species in the interface [22]. Thus, both impedance spectra have been analyzed using an equivalent circuit, Figure 2. In this circuit, R_s_ represents the resistance of the electrolyte, *W* is the Warburg diffusion, and *CPE* is the non-ideal double layer capacitance, which is defined by two parameters, *Y*_0_ and *n*, by Equation (1):(1)$$\mathrm{CPE}=\frac{1}{\left[Y_{0}{\left(j\omega \right)}^{n}\right]}$$
where *ω* is the angular frequency, and *j* = √−1.

A good fit of the circuit can be confirmed in Figure 3. The fitted parameters are shown in Table 1. Similar values of R_s_ and W can be observed for both cases. This agrees with the expected results because R_s_ depends mainly on the solution conductivity, which remains almost the same in both cases, and W depends on the diffusion coefficient and probe concentration, which do not change. In contrast, significant differences are observed in R_tc_ and *Y*_0_. The lower value of *Y*_0_ = 0.304 (µF)*^n^* for the modified electrode compared to the bare electrode, *Y*_0_ = 3.04 (µF)*^n^*, is a consequence of an increase in the electrical insulating character of the electrode interface due to the presence of the boron cage and its enhanced capacity to store charge. The double layer capacitance for the modified electrode is also higher (15.6 µF/cm^2^) than those reported for a SAM with Au with mercaptoundecanoic acid (MUA) and mercaptoacetic acid (MAA) under similar conditions, 1.50 µF/cm^2^ and 0.290 µF/cm^2^, respectively [23]. This effect is attributed to the large value of the dipole moment (µ) of [B_12_H_11_SH]^2−^ compared to the alkanethiols mentioned above. This effect could explain the high stability of the gold cluster Au_55_[Na_2_B_12_H_11_SH]Cl_6_ due to the presence of the boron cage observed only by transmission electron microscopy (TEM), reported in literature [10]. The fraction of [B_12_H_11_SH]^2−^ coating on the Au surface can be calculated by the ratio of charge transfer resistances in absence (*Rct*^0^) and in the presence (*Rct*) of BSH, according to Equation (2).
(2)$$\left(1-\theta \right)=\frac{R_{ct}{}^{0}}{R_{ct}}$$

In this case, 92.2% of the Au surface is occupied by the BSH molecules. This high value can be explained in terms of bulky effects of the boron cage, which hinders the presence of the probe Fe(CN)_6_^3−^/Fe(CN)_6_^4−^ at the electrode interface. The same electrochemical experiments using an Au electrode modified with thioglycolic acid (TGA) were caried out. For this assembly, 82% of the Au surface is occupied by TGA, suggesting that the boron cage [B_12_H_11_SH]^2−^ diminishes the actives sites over the Au surface due to the steric effects, (see Figure 3).

At this point, it was demonstrated that the boron cage [B_12_H_11_SH]^2−^ modified a polycrystalline gold electrode. Considering the capacitance measured as two parallel capacitors, the dielectric constant of the boron cage SAM (εBSH) and the thickness of the layer, see Equation (3), can be estimated. Therefore, DFT calculations and surface plasmon resonance experiments were carried out to obtain the thickness of the layer.
(3)$$C_{dl}=\frac{\varepsilon_{0}\varepsilon_{BSH}}{d}$$

### 3.3. Theoretical Calculations for [B_12_H_11_SH]^2−^

DFT calculations of the anion [B_12_H_11_S-Au]^2−^ were performed. The optimized geometry is presented in Figure 4A, where the calculated distance between the sulfur atom and the hydrogen located in the apical position of the boron cage shows a value of 6.531 Å. A molar volume of 167.645 cm^3^/mol was also calculated. Figure 4B presents the highest occupied molecular orbital (HOMO) for the anion [B_12_H_11_S-Au]^2−^, in which it can be observed that the electron density is mostly located on the sulfur atom, indicating the site with the highest affinity to form bonds or attach to the gold surface.

In addition, chemical descriptors were calculated to provide support to the theoretical calculations for the anion [B_12_H_11_S-Au]^2−^ and for the thioglycolic acid (TGA) (see Table 2). The chemical descriptors employed were chemical hardness (*η*), chemical potential governing charge donating process (*μ*^−^), chemical potential governing charge accepting process (*μ*^+^), electrodonating power (*ω*^−^), and electroaccepting power (*ω*+), which depends on the ionization energy (*I*) and electron affinity (*A*), as seen in Equations (4)–(8) [24]. Similar values for the descriptors were obtained for both species, which indicate that the reactivity of both molecules are centered over sulfur atoms. Conversely, a high value of dipole moment was obtained for [B_12_H_11_SH]^2−^. This fact suggests a high dielectric character for the boron cage.
(4)$$\eta =\frac{1}{2}\left(I-A\right)$$
(5)$$\mu^{-}=\frac{-1}{4}\left(3I-A\right)$$
(6)$$\mu^{+}=\frac{-1}{4}\left(I-3A\right)$$
(7)$$\varpi^{-}\equiv \frac{{\left(\mu^{-}\right)}^{2}}{2\eta }\approx \frac{{\left(3I+A\right)}^{2}}{16\left(I-A\right)}$$
(8)$$\varpi^{+}\equiv \frac{{\left(\mu^{+}\right)}^{2}}{2\eta }\approx \frac{{\left(I+3A\right)}^{2}}{16\left(I-A\right)}$$

### 3.4. Optical Thickness by Surface Plasmon Resonance

Figure 5 shows the SPR angular scan curves for the untreated Au surface and for the [B_12_H_11_S-Au]^2−^ SAM prepared by adsorption from an aqueous solution for 90 min (Figure 4). From this result, it was found that the position of the *θ*_SPR_ angle of the sensor chip surface was shifted to a higher angle of incidence due to adsorption of the SAM on gold. The best fit, indicated by the solid lines in the figure, give an optical thickness of the SAM *d* = 9 Å, considering a refractive index *n* = 1.50. This thickness is slightly different to the length of the molecule (6.531 Å), calculated from the geometry optimization of the anion [B_12_H_11_SH]^2−^ from DFT calculations. The difference between these values can be attributed to solvation effects on the boron cage.

### 3.5. Evaluation of the Dielectric Constant of the [B_12_H_11_S-Au]^2−^ Monolayer

As it was mentioned before, for a modified electrode, the total capacitance can be considered as two parallel capacitors, one for pinholes and the other related to the monolayer. Thus, the capacitance of the SAM gold electrode can be described with Equation (9) [23]:(9)$$C=C_{dl}^{0}\left(1-\theta \right)+C_{m}^{0}\theta$$

The *C*^0^*_dl_* values correspond to the capacitance of the unmodified electrode, *C*^0^*_m_* is the monolayer capacitance assuming an ideal 100% coverage. *θ* is the experimental coverage on the modified electrode, obtained from EIS experiments. The dielectric constant of the boron cage SAM (*ε_BSH_*) can be calculated according to Equation (10) [23], where the *d* value corresponds to the monolayer thickness. ε_o_ corresponds to the permeability at vacuum (8.85 × 10^−12^ C^2^N^2^m^2^).
(10)$$\varepsilon_{BSH}=\frac{C_{m}^{0}d}{\varepsilon_{0}}$$

With the value of the optical thickness of the SAM, obtained from SPR experiments, a dielectric constant value *ε_BSH_* of 0.224 was calculated, which is lower than the values reported for a SAM of 3-mercaptopropionic acid (MPA) and mercaptoundecanoic acid (MUA) under similar conditions, 0.677 and 0.637, respectively.

### 3.6. Differential Capacitance Studies of the [B_12_H_11_S-Au]^2−^ Monolayer

Differential capacitance curves of the boron cage monolayer, TGA monolayer and bare gold electrode were obtained. For gold electrode curve (Figure 6a), a peak at 250 mV vs. Ag/AgCl is observed, which can be identified as a reconstruction of the Au (111) surface [25]. The formation of [B_12_H_11_S-Au]^2−^ and TGA monolayers can be evidence due to the absence of a gold reconstruction peak and a decrease in capacitance values with respect to the gold electrode. When the [B_12_H_11_S-Au]^2−^ monolayer is formed, no peaks can be observed (Figure 6b), indicating that [B_12_H_11_S]^2−^ adsorption prevents the reconstruction Au(111) processes. The C_d_ values for [B_12_H_11_S-Au]^2−^ present no dependence with the applied potential, resembling ideal polarizable electrode behavior. In this approach, the double layer can be described as a parallel plate capacitor, already mentioned, where the C_d_ values are defined by the [B_12_H_11_S]^2−^ dielectric constant and the formed monolayer uniformity and thickness. In the same way, when the gold electrode is modified with the TGA monolayer, there is a decrease in C_d_ values, as seen in Figure 6c. However, the monolayer formed by TGA presents defects that allow got the gold reconstruction process with a displacement in the potential that occurs at 0 mV vs. Ag/AgCl. Differential capacitance results for modified gold electrodes with [B_12_H_11_SH]^2−^ and gold (TGA) indicate that bulky thiols enhance charge accumulation at the electrode–solution interface.

### 3.7. Adsorption Kinetics by Surface Plasmon Resonance of the [B_12_H_11_S-Au]^2−^ Monolayer

Time-depending monitoring of the SPR response could be used for direct visual observation of binding and dissociation events during the reaction. Real-time analysis is an ideal way to obtain kinetic data. In addition, interaction monitoring often provides valuable dynamic information. In this study, we used the real-time NanoSPR technique to monitor the self-assembling process, as shown in Figure 7. This figure shows that the injection of the analyte containing [B_12_H_11_SH]^2−^ caused an increase in the SPR response of about 0.12°. Since the increase in the *θ*_SPR_ response implies the increase in the analyte mass on the surface of the sensor chip, it was concluded that a self-assembled monolayer of [B_12_H_11_SH]^2−^. was formed on the gold-coated glass slide. At the end of the adsorption process, the chip surface was rinsed with water. The thin arrows in Figure 7 indicate sample injection (↑) and a rinsing step (↓). The kinetic sensorgram (Figure 5) indicates that the adsorption of [B_12_H_11_SH]^2−^. can be modeled using a Langmuir kinetic model; see Equation (11) [26]:(11)$$\theta_{t}\;=\;\left(\theta_{max}-\theta_{0})\left[1-\exp(-k_{obs}t\right]\right)+\theta_{0}$$
where *θ_t_* represents the change in *θ*_SPR_ unit response at time t, *θ_max_* represents the maximum *θ*_SPR_ unit response, *θ*_0_ is the unit response before injection of BSH, and *k_obs_* is the observed “rate constant”.

We can calculate the observed rate constant for the adsorption of BSH on the sensor chip surface by fitting Equation (11) to the kinetic sensorgram in Figure 7. For this sensorgram, *k_obs_* = (5.01 ± 0.06) × 10^−2^ s^−1^ (R^2^ = 0.9428).

## 4. Conclusions

By means of cyclic voltammetry, an interaction between the -SH from the boron cage, and Au(OH)_3_ ↓ from the gold electrode, as a first step for the electrode modification was demonstrated.

Changes in the cyclic voltammetry and electrochemical impedance spectroscopy for the probe Fe(CN)_6_^3−^/Fe(CN)_6_^4−^ allowed us to establish the formation of a [B_12_H_11_S-Au]^2−^ SAM over a polycrystalline electrode.

Electrochemical experiments indicate that the boron cage diminishes the actives sites over the Au surface due to the steric effects, with 92.2% of the coating over the electrode.

The obtained value of double layer capacitance for the modified electrode with the boron cage (15.6 µF/cm^2^) is characteristic of a higher dipole moment (*µ*) in comparison with other alkanethiols such as mercaptoundecanoic acid (MUA) and mercaptoacetic acid (MAA).

Differential capacitance results indicate that SAM over gold electrodes with bulky thiols such as [B_12_H_11_SH]^2−^ enhance charge accumulation at the electrode–solution interface.

It was observed that C_d_ values for [B_12_H_11_S-Au]^2−^ SAM presented a constant value no matter the applied potential.

Electrochemical experiments and surface plasmon resonance measurements (SPR) allowed us to calculate a molecular length value 9 Å and the dielectric constant of the Boron cage in the SAM of 0.224.

The HOMO for the [B_12_H_11_SH]^2−^ is located on the sulfur atom, indicating the site with the highest affinity to form bonds or attach to the gold surface.

The similar values of the chemical descriptors, chemical hardness (*η*), chemical potential governing charge donating process (*μ*^−^), chemical potential governing charge accepting process (*μ*^+^), electrodonating power (*ω*^−^), and electroaccepting power (*ω*^+^) for [B_12_H_11_SH]^2−^ and TGA suggest that the reactivity of both molecules is centered over sulfur atoms.

From DFT calculations, a high value of dipole moment was obtained for [B_12_H_11_SH]^2^, which indicates a high dielectric character.

The SPR kinetic sensorgram indicates that the adsorption of the boron cage obeys the Langmuir kinetic model *k_obs_* = (5.01 ± 0.06) × 10^−2^ s^−1^ typical for the formation of a monolayer.

## Data Availability

No supporting information is included.

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
