# Peer review of "Self-Assembled Monolayer of Monomercaptoundecahydro-closo-dodecaborate on a Polycrystalline Gold Surface"

_molecules, 2022, doi:10.3390/molecules27082496_

Round 1

Reviewer 1 Report

The manuscript deals with nano-layers developed for electrochemistry. As such, it fits in with the subject of the journal. However, it requires corrections and supplements to verify the novelty of the results and the consistency of the manuscript. Detailed comments are provided below.
1. The Conlusions section is very short. The description of achievements should be extended and related to the state of art.
2. In the Introduction, the authors refer to cancer therapy. Later in the article, this problem is absent, both in the design of the experiment, as well as in the analyzes and measurements. There is also no reference to cancer in Conclusions. The introduction is thus inconsistent with the rest of the article. Significant changes need to be made (removed or referenced) and corrected.
3. What's new in the article? Measurements for Fe (CN) 6
3− / Fe (CN) 6 4− are routine in electrochemistry. Were any unique properties or parameters obtained?
4. There is no name of the country in the 1. affiliation.

Author Response

Dear Reviewer

All the points raised by you, were answered. We do hope the corrections made improve the quality of the work and make it suitable for publication in Molecules.

Please see the attachment file

Reviewer 2 Report

This study invesitigates different aspects of the SAM on Au surface formed by adsorption of mercapto-appended boron cluster. The authors employed electrochemical, time-independent and time-dependent SPR angle measurement as well as computational simulation to establish the thickness, coverage, change in dielectric constant and molecular orientation of the SAM. Overall I found the experimental design and analysis to be sound and well-articulated. Although this research lacks significant novelty and the manuscript resembles a lab report, the readers would still benefit from learning how the authors combine different approaches to study a well-defined system. I therefore recommend this manuscript for publication.

Minor comments:

  1. I found the "Boron neutron capture therapy (BNCT)” part of the introduction paragraph largely irrelevant to the topic of this study. The authors could discuss the background of SAM on Au.
  2. Section 3.5 should follow section 3.2 because it uses some of the derived values from 3.2

Author Response

(The authors gave the same response as above.)

Reviewer 3 Report

All comments are listed in the paper.

The introduction is rather scantily written.

It is necessary to search the literature in more detail

It is necessary to try to determine the thickness of the layer with EIS! 

There is almost no discussion!

No scheme or equation is shown, i.e. it is necessary to assume the self-assembling mechanism.
Much more analysis of the modified surface, as well as more detailed DFT calculations, is needed.

Author Response

(The authors gave the same response as above.)

Reviewer 4 Report

The work presents the characterization of a self-assembled monolayer for the boron 16 cage [B12H11SH]2-, on a polycrystalline gold electrode. The work is well written and the manuscript could be considered for publication in “molecules” after considering the following points:

  • In the whole manuscript, the subscript numbers in the chemical formula and the superscript numbers for the charge are not justified.
  • In section 2.2., there is no carbon in the chemical formula but the elemental analysis showed result for the carbon. I think this should be hydrogen.
  • Please, check the font of the text at the end of section 2.3. The same for section 2.4.
  • In section 2.6, All quantum mechanics calculations……should be…. All quantum chemical calculations
  • Please check this sentence: “the distance measured between the 265 sulfur atom and the farthest hydrogen was 6.531 Å-------“ the word measured should be replaced by calculated.
  • Also, there is S-H bond and there is a hydrogen atom bonded to sulphur which is the closest hydrogen..is it possible that this distance is 6.531 Å??

Author Response

(The authors gave the same response as above.)

Round 2

Reviewer 1 Report

The manuscript deals with nano-layers developed for electrochemistry. As such, it fits in with the subject of the journal. The Authors introduced corrections suggested in previous review. The area is known and Authors used known materials for test however the paper can be published as new material was used for electrode.

Reviewer 3 Report

I appreciate everything you did but you could also try to determine the thickness with EIS.